# DNA Protection Protein, a Novel Mechanism of Radiation Tolerance: Lessons from Tardigrades

**DOI:** 10.3390/life7020026

**Published:** 2017-06-15

**Authors:** Takuma Hashimoto, Takekazu Kunieda

**Affiliations:** 1Laboratory for Radiation Biology, School of Medicine, Tohoku University, 2-1 Seiryo-machi, Aoba-ku, Sendai, Miyagi 980-8575, Japan; t.hashimoto@med.tohoku.ac.jp; 2Department of Biological Sciences, Graduate School of Science, The University of Tokyo, 7-3-1 Hongo, Bunkyo-ku, Tokyo 113-0033, Japan

**Keywords:** tardigrade, extremophiles, radiotolerance, damage suppressor (Dsup), reactive oxygen species (ROS)

## Abstract

Genomic DNA stores all genetic information and is indispensable for maintenance of normal cellular activity and propagation. Radiation causes severe DNA lesions, including double-strand breaks, and leads to genome instability and even lethality. Regardless of the toxicity of radiation, some organisms exhibit extraordinary tolerance against radiation. These organisms are supposed to possess special mechanisms to mitigate radiation-induced DNA damages. Extensive study using radiotolerant bacteria suggested that effective protection of proteins and enhanced DNA repair system play important roles in tolerability against high-dose radiation. Recent studies using an extremotolerant animal, the tardigrade, provides new evidence that a tardigrade-unique DNA-associating protein, termed Dsup, suppresses the occurrence of DNA breaks by radiation in human-cultured cells. In this review, we provide a brief summary of the current knowledge on extremely radiotolerant animals, and present novel insights from the tardigrade research, which expand our understanding on molecular mechanism of exceptional radio-tolerability.

## 1. Tardigrades as Model Animals Tolerant to Various Extreme Environments

“Extremophiles” are organisms that have adapted or at least are tolerant to extremely harsh environments. Most extremophiles are single cellular organisms with simple structure, such as archaea and bacteria. However, some animals also exhibit extraordinary tolerance against extreme environments [1]. One of the most known examples of such extremotolerant animals is the tardigrade. Tardigrades are aquatic invertebrates discovered by German zoologist Goeze in 1773. Their body is typically 0.1–1.2 mm length (Figure 1) and is composed of five segments; one head segment and four trunk segments with a pair of legs each [2]. They are in microscopic scale, but possess a well-developed nervous system including brain [2]. Because of their slow movement, the organisms were termed Tardigrada, from Latin tardigradus, slow-stepper, by Italian natural scientist Spallanzani in 1776 [3]. Currently, approximately 1200 species have been reported from various habitats, such as marine, fresh-water, or limno-terrestrial environments, though the real number of tardigrade species is estimated to be much higher [4,5]. Tardigrades were found in harsh environments such as Antarctica as well as in urban area, e.g., from activated sludge in a sewage treatment plant [6,7,8]. All tardigrades require surrounding water to grow and reproduce, but some limno-terrestrial species are able to tolerate almost complete dehydration. When the surrounding water evaporates, tolerant tardigrades lose almost all body water and enter a metabolically inactive dehydrated state called anhydrobiosis [9]. The dehydrated tardigrades withstand various extreme conditions that normally disallow the survival of most other organisms; for example, low and high temperatures (from −273 °C [10] to nearly 100 °C [11,12]), high hydrostatic pressure (7.5 GPa) [13], immersion in organic solvent [11,14], and exposure to a high dose of irradiation [11,15,16]. Furthermore, tardigrades are the first animals to have revived after exposure to outer space for 10 days [17]. The molecular mechanisms enabling these exceptional resistances, however, are not well understood.

Recently, the genetic information of tardigrades is rapidly expanding, and the genome sequences have been determined for two tardigrade species: an extremotolerant terrestrial species, *Ramazzottius variornatus*, and a less tolerant freshwater species, *Hypsibius dujardini* [18,19,20]. The decoded genome of *R. varieornatus* has much better contiguity and coverage (the span is 55.8 Mbp in only 199 scaffolds; N50 = 4.74 Mbp; 96.8% coverage of core eukaryotic gene set) than the other, and is thus considered as a reference genome of the phylum Tardigrada. The high-quality genome of the extremotolerant tardigrade provides a solid foundation for molecular dissection of tardigrade tolerant ability. Indeed, using the genome information, several desiccation-tolerance-related genes have been identified, such as CAHS (cytoplasmic abundant heat soluble), SAHS (secretory abundant heat soluble), MAHS (mitochondrial abundant heat-soluble), and LEA (late embryogenesis abundant) proteins [21,22,23]. Recent rapid progresses of molecular analyses make tardigrades an attractive animal model in extremophile research.

## 2. Radiotolerant Organisms

Proper maintenance of genomic DNA is important for preserving correct genetic information and normal cellular functions. Genomic DNA is constantly exposed to various genotoxic stresses of both intrinsic and extrinsic origin, e.g., metabolically generated reactive oxygen species (ROS) or radiation exposure [24,25]. Organisms employ multiple mechanisms to combat these damaging agents, including various DNA repair processes and the detoxification of the causative agents. In most organisms, however, the tolerable capacity is limited, and excessive genotoxic stress such as high-dose radiation, causes severe DNA lesions, including double-strand breaks (DSBs), and leads to genome instability and less viability [26,27,28]. This limitation prevents these organisms from advancing to genotoxic environments; e.g., space radiation is considered a major hazard for humans traveling outer space [29].

For mammals, doses of several Gray (Gy) units of ionizing radiation are fatal. The median lethal dose (LD_50_) is approximately 7 Gy for mouse and approximately 4 Gy for human [30]. In general, prokaryotes exhibit much higher radiation tolerance than animals. For instance, the LD_50_ of *Escherichia coli* is in a range from 50 to several hundred Gy depending on a strain. However, almost all *E. coli* perish when they are irradiated with 2000 Gy [31]. On the other hand, *Deinococcus radiodurans*, one of the most famous radiotolerant bacteria, is reported to survive without loss of viability even after exposure to irradiation with 5000 Gy of gamma-rays [32,33]. Irradiated *D. radiodurans* suffer severe fragmentation of its genomic DNA, but their fragmented DNA is rapidly patched up to complete circular genome by extensive DNA repair system [34]. Extensive studies revealed that the radiotolerant bacteria utilize three major strategies contributing to its radiation tolerance: antioxidant defenses, cellular cleaning, and DNA repair [35,36,37,38,39]. Antioxidant defenses prevent radiation induced damage to their proteomes and protect DNA repair enzymes. This enables the effective repair of DNA damage using the polyploid genome, and the elimination of severely damaged molecules to recover cellular integrity. Mutation of DNA repair pathway compromises the radiotolerance of *D. radiodurans*, suggesting that the high radioresistance depends in part on a powerful DNA repair system [39].

In animals, bdelloid rotifers and larvae of *Polypedilum vanderplanki* (sleeping chironomid) have the ability to enter anhydrobiosis, and in a dehydrated state they can withstand several thousand Gy of gamma irradiation [40,41]. Both animals exhibit fragmentation of DNA by high-dose irradiation with a level similar to other ordinary animals. Therefore, it is suggested that their high radiotolerant abilities also depend on the DNA repair system. As rotifers possess a high antioxidant capacity, antioxidant defenses and DNA repair are proposed to contribute to radiotolerance in animals as well [42]. Their radiotolerances are significantly lower in a hydrated state. For instance, approximately 80% of *P. vanderplanki* larvae in a dehydrated state survive after 4000 Gy of gamma irradiation, but only 30% larvae survive in a hydrated state. It is well known that approximately two-thirds of the X-ray damage to DNA in mammalian cells is caused by hydroxyl radicals generated from irradiated water molecules (indirect effect [43,44]). Thus, it is assumed that dehydrated animals are expected to be less damaged by radiation and exhibit better tolerance compared to hydrated conditions [45].

It is known that the sensitivity to irradiation is affected by the change of the chromatin structure. Indeed, chromatin relaxation and the loss of the chromatin protein increase the number of DNA DSBs after irradiation [46]. In radiotolerant organisms, it is suggested that DNA condensation may contribute to the radiotolerance. Indeed, a condensed ring-like structure of nucleoid was observed in *D. radiodurans* [47], and comparative analyses among several species in *Deinococcaceae* suggested that the extremely radioresistant species exhibited more condensed genome structures than those in the radiosensitive species [48]. The tight and ordered DNA packaging might be proposed to facilitate DNA repair by promoting both template-independent DNA joining and RecA-dependent recombination.

## 3. Extraordinary Tolerance to Irradiation in Tardigrades

Certain tardigrade species can tolerate high doses of ionizing radiation. Current knowledge about radiotolerances of tardigrades is summarized in Table 1. Typically, limno-terrestrial species, e.g., *R. varieornatus*, *Milnesium tardigradum, H. dujardini*, and *Richtersius coronifer*, were able to withstand several thousand Gy of radiation [11,15,16,49].

It is noteworthy that the periods used for LD_50_ estimation are different among analyses. The regular period for LD_50_ estimation is 60 days for humans and is 30 days for mice, whereas the most analyses using tardigrades and other invertebrates, e.g., *P. vanderplanki*, are relatively much shorter, ranging from 24 h to 7 days. This could be justified due to the differences in life span (usually one to several months in tardigrades) and genome size among animal species, but the used duration should be taken into account when LD_50_ values are compared among different animals. In humans, doses higher than 50 Gy cause convulsions over the entire body and severe shock, which leads to death within five days. In the case of mice irradiated with 800–1000 Gy, they can survive for only 0.1–0.01 day [50]. Thus, the radiotolerance of tardigrades, which can survive for several days after 4000 Gy of irradiation, should be considered extraordinary. Furthermore, some tardigrade eggs were able to hatch even after irradiated with 2000 Gy of ^4^He ions in dehydrated condition, and LD_50_ values were reported as 1690 Gy for dehydrated eggs and 509 Gy for hydrated ones [51], indicating that tardigrades have a high radiotolerance even in early embryonic stages.

One of the characteristic features of the radiotolerance of tardigrades is that they exhibit high radiotolerance, even in a hydrated state with a level similar to those in a dehydrated state [11], whereas other radiotolerant animals exhibit certainly more tolerant in a dehydrated state. There is another different feature in tolerability between tardigrades and other desiccation-tolerant animals. Dehydration usually causes DNA damage as radiation does. In sleeping chironomid, dehydration made 40–50% of DNA fragmented, and this is comparable to DNA damage induced by irradiation with a 70-Gy He ion-beam [52]. In contrast, a desiccation-tolerant tardigrade, *M. tardigradum*, exhibited much fewer DNA breaks after dehydration stress, with only approximately 2% of DNA detected as fragmented after 2 days of dehydration [53]. DNA in tardigrades seems to suffer much less damage by dehydration stress compared to other animals [53,54]. These observations lead to the postulation that tardigrades possess the mechanism protecting DNA from damaging agents and that such mechanisms contribute to the extreme radio-tolerance of the animal. Recent studies have identified a tardigrade-unique DNA-associating protein, Dsup, as a potential DNA protectant in tardigrades [18]. This novel finding and its meanings will be discussed in the following sections.

## 4. Tardigrade-Unique DNA-Associated Protein, Dsup, Improves Radiotolerance

*R. varieornatus* is one of the most radiotolerant species in tardigrades [11]. Considering DNA as a major target of radiation damage, the tardigrade is assumed to possess some proteins associated with DNA to protect and/or to effectively repair DNA. Recently, as a representative of such proteins, Damage suppressor (Dsup) was identified from a chromatin fraction of the tardigrade [18]. A human cultured cell line was engineered to express Dsup protein and was irradiated with X-rays. Intriguingly, such engineered cells exhibited substantially suppressed (approximately half) DNA fragmentation compared to non-engineered cells. In the analyses, cells were irradiated on ice and DNA fragmentation was detected immediately after irradiation, suggesting that the suppressed fragmentation was detected before significant DNA repair occurred. Thus, the reduced DNA fragmentation in Dsup-expressing cells was likely due to the reduced occurrence of DNA breaks rather than facilitation of the DNA repair process. This notion was further evidenced by the fact that Dsup-expressing cells exhibited a much lower number of the DNA break marker γ-H2AX, which accumulates shortly after irradiation and is usually retained for several hours, even after completion of DNA break repair (Figure 2A) [59,60,61,62]. In addition to irradiation stress, Dsup-expressing cells exhibited a significant reduction in DNA fragmentation when exposed to hydrogen peroxide (a kind of ROS), compared to untransfected cells [18]. Thus, Dsup protein has ability to protect DNA from ROS and serves as a DNA protectant against indirect effects of X-rays.

Irradiation with a sublethal dose (e.g., 4 Gy) of X-rays leads to a loss of proliferative ability in mammalian cells [63]. Surprisingly, many Dsup-expressing cells exhibited a normal morphology even after irradiated with 4 Gy of X-ray, and the cell numbers increased over time, suggesting that these irradiated cells retained proliferative ability (Figure 2B). Considering these results, Dsup protein is able to confer not only DNA protection but also improved radiotolerance to human cultured cells. Enhanced DNA repair has been supposed to be an important basis for the high radiotolerance, but DNA protection might also be another key factor, at least in tardigrade’s resilience.

## 5. DNA-Association is Necessary for DNA Protection Activity of Dsup Protein

C-terminal region of Dsup is required and sufficient for association with DNA in vitro and for colocalization with nuclear DNA in transfected cells. A stable line expressing a mutant Dsup protein, Dsup∆C, which lacks the C-terminal DNA-associating region, exhibited no reduction in DNA fragmentation compared to control cells. Therefore, the association with DNA is prerequisite for Dsup protein to protect DNA from X-rays. On the other hand, transient expression of C-terminal region of Dsup alone (Dsup-C) co-localized with nuclear DNA, but induced an abnormal aggregation of nuclear DNA, and we could not establish any stable cell liens expressing Dsup-C. It is possible that nonspecific binding of Dsup-C to DNA interferes with DNA replication and/or transcription, thereby preventing cell proliferation. Similar adverse effects were reported for some DNA-binding proteins; e.g., overexpression of a bacterial histone-like nucleoid-structuring (H-NS) protein, or a small acid-soluble spore protein (SASP) associated with spore DNA of *Bacillus subtilis*, causes severe condensation of DNA and loss of cell viability [64,65]. In contrast, full-length Dsup-expressing cells exhibited an almost normal distribution of nuclear DNA, similar to that in control cells. The N-terminal region and the predicted α-helical region at the middle (Figure 3) would be important to relieve the adverse effects induced by the association of proteins with DNA (e.g., possible heterochromatinization and/or interference on transcription and replication).

## 6. What is the Origin of Dsup, a DNA Protective Protein?

Dsup protein has a characteristic amino acid sequence, and no similar proteins have been retrieved using a BLAST search against NCBI non-redundant database. Thus, we carefully searched for potential Dsup homologues from tardigrade sequence databases individually and identified a protein exhibiting weak similarity with Dsup (bit-score = 34.3; *E*-value = 0.09) from the recent predicted proteome of a freshwater tardigrade, *H. dujardini* [20]. The protein is annotated as ‘hypothetical protein BV898_01301’ in the NCBI database, and no functional information is available. This protein is composed of 328 amino acids, which is slightly shorter than that of the original Dsup protein of *R. varieornatus* (445 amino acids). Pairwise alignment using MAFFT revealed a certain similarity between two proteins with 26.4% identity and 35.5% similarity. Subcellular localization prediction using WoLF PSORT suggested nuclear localization for both proteins, and a putative nuclear localization signal is predicted in both proteins at a similar position near C-terminus by using cNLS Mapper software (Figure 4A). Despite weak similarity in primary structure between two proteins, both proteins exhibited similar profiles in hydrophobicity and charge distribution along protein. Both protein has a relatively broad hydrophobic region at the middle position, and C-terminal halves of both proteins are positively charged showing a characteristic charge distribution pattern well-conserved in two proteins (Figure 4B). Based on the observed similarity between two proteins, e.g., certain similarity in the primary structure, the position of NLS and profiles of hydrophobicity and charge distribution, we consider this protein as a potential Dsup orthologue in *H. dujradini*. Two species, *R. varieornatus* and *H. dujardini,* belong to the same taxonomic family Hypsibiidae, but the protein sequences of Dsup protein are unexpectedly diverged between two species. This suggests that the primary structure of Dsup has been under weak selective pressure during evolution. Similar sequence diversifications are observed in some intrinsically disordered proteins, because amino acids in the disordered region are generally rather changeable. For example, late embryogenesis abundant (LEA) proteins, which work as an unstructured desiccation protectant, exhibit high diversification in protein sequences and their flexible structures are proposed to function as a physical shield protecting other biomolecules or as a material supporting glass transition during desiccation [66,67,68]. Accordingly, we speculate that Dsup protein might function with a flexible structure rather than in a rigid form, e.g., as a physical shield of DNA rather than as an enzyme. Although overall sequence similarity is relatively low in Dsup proteins, some short motifs are strictly conserved between two proteins, e.g., KEKSKSPAKEV at positions 90–100 or AKGRGXRGRXPAAXK at positions 275–289. These motifs could be important for Dsup function. Future mutational analyses will reveal the importance of these conserved motifs.

## 7. Concluding Remarks

In this review, we summarized the current knowledge of extremely radiotolerant animals, mainly focusing on tardigrades as an emerging animal model of extremophiles. Antioxidant defenses and the efficient DNA repair by the protected enzymes have been accepted as a common basis for elevated radiotolerance shared from prokaryotes to animals. The recent genome analysis revealed that tardigrades also possess redundant copies of antioxidant enzymes and DNA repair enzymes, while lacking ROS-producing enzymes [18], so a similar principle could be applicable to tardigrades as well. The recent finding of the novel DNA protection protein in tardigrades provides strong evidence that DNA protection could also be a possible mechanism contributing extraordinary radiotolerance in addition to the well-accepted “protection of protein and repair of DNA” principle.

Although the precise mechanisms of DNA protection by Dsup protein remain to be elucidated, the association with DNA is important for protection activity of Dsup protein, suggesting a possible physical shielding of DNA from ROS and irradiation, and/or a local detoxification of ROS as potential mechanisms (Figure 5). Considering that Dsup is a novel protein unique to tardigrade, tardigrades could have invented their own tolerant mechanisms during evolution. Genome analysis of *R. varieornatus* revealed the presence and abundant expressions of many tardigrade-unique genes, so tardigrades and potentially other extremotolerant animals could be a bountiful resource of unidentified tolerance genes and mechanisms. Recent rapid progress of molecular analyses of tardigrades and other extremotolerant animals should accelerate elucidation of novel tolerant mechanisms and expand our understanding of the molecular mechanisms of extremotolerance. Intriguingly, the Dsup story tells us that the tolerant ability of extremotolerant animals could be transferred to more sensitive organisms at least partly by transferring the corresponding genes. Unveiling the molecular mechanisms underlying extremotolerance in tardigrades will provide novel clues that open new avenues to confer stress resistance to intolerant species, including humans.

## Figures and Tables

**Figure 1 life-07-00026-f001:**
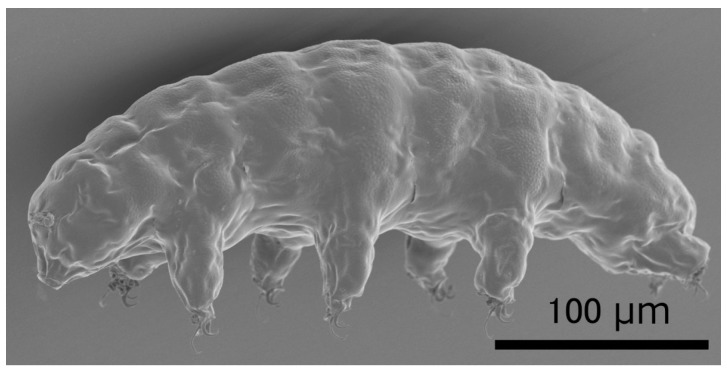
Scanning electron microscopy images of the extremotolerant tardigrade, *R. varieornatus*. Reproduced from [18].

**Figure 2 life-07-00026-f002:**
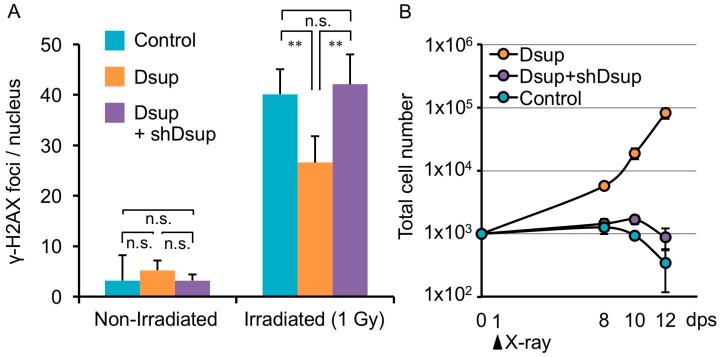
Dsup reduced X-ray-induced DNA damage and improved viability of irradiated human cultured cells. (**A**) Quantitative comparison of DNA-break marker, γ-H2AX foci number among untransfected human cultured cells (Control), Dsup-expressing cells (Dsup), and Dsup-knockdown cells (Dsup + shDsup) under non-irradiated and 1 Gy X-ray irradiated conditions. ** *p* < 0.01, n.s. indicates not significant (Tukey–Kramer’s test). (**B**) Comparison of growth curves of untransfected cells (Control), Dsup-expressing cells (Dsup), and Dsup-knockdown cells (Dsup + shDsup) in irradiated conditions. Values represent mean ± s.d. Reproduced from [18].

**Figure 3 life-07-00026-f003:**
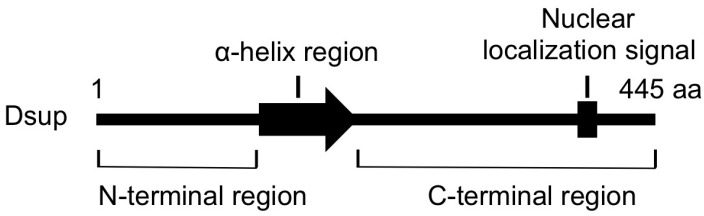
The schematic structure of Dsup protein. This protein is composed of 445 amino acids (aa). The arrow indicates a predicted α-helix at the middle region and a bar in the C-terminus indicates a predicted nuclear localization signal.

**Figure 4 life-07-00026-f004:**
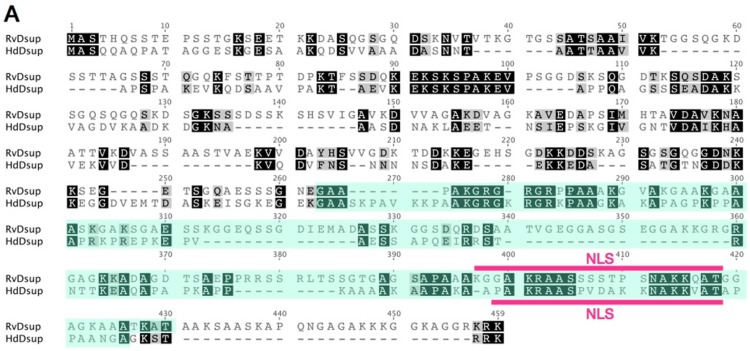
Comparison of putative Dsup proteins in two tardigrade species. (**A**) Pairwise alignment of Dsup protein of *R. varieornatus* (Rv_Dsup; accession number = BAV59442) and putative Dsup protein of *H. dujardini* (Hd_Dsup, hypothetical protein BV898_01301, accession number = OQV24709). Identical residues and similar residues are shown in inverted boxes and shaded boxes, respectively. Predicted nuclear localization signals (NLS) are shown by red bars. Green shades indicate conserved alanine-rich regions detected by PROSITE protein pattern search. (**B**) Comparison of distribution profiles of hydrophobicity and charges between two tardigrade Dsup proteins. Hydrophobicity and charge distribution were analyzed using the ProtScale program at ExPASy and EMBOSS charge program, respectively.

**Figure 5 life-07-00026-f005:**
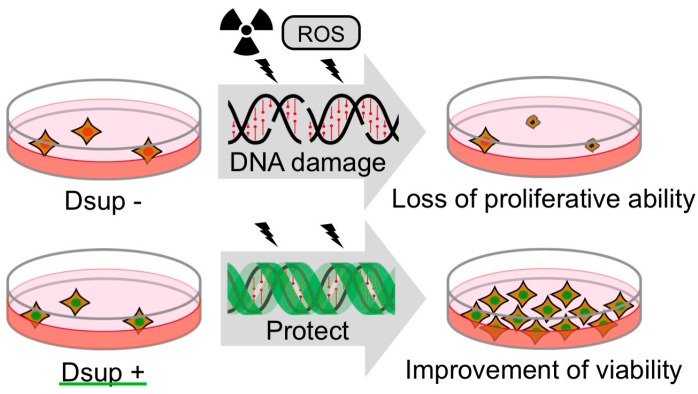
Schematic model of DNA protection by Dsup protein from radiation damage. Radiation induces DNA breaks, which could interfere DNA replication and gene expression. Heavily damaged cells lose their proliferative ability and are destined for death. Dsup protein suppresses X-ray induced DNA damage depending on association with nuclear DNA, possibly through physical shielding from, or detoxification of, reactive oxygen species (ROS) generated through indirect radiation effects. Thereby, Dsup protein can improve the radiotolerance of cultured animal cells.

**Table 1 life-07-00026-t001:** Tolerance to ionizing radiation in tardigrades.

Tardigrade Species	State ^1^	Ionizing Radiation	Radiotolerance	Reference
*Macrobiotus areolatus*	Dehyd.	X-ray	5700 Gy (LD_50/1 day after_)	[55]
*Richtersius coronifer*	Hyd.	γ-rays	4700 Gy (LD_50/18 h_)	[16]
	Hyd.	γ-rays	2500 Gy (LD_50/30 days_)	[16]
	Dehyd.	γ-rays	3000 Gy (LD_50/22 h_)	[16]
	Dehyd.	proton-beam	10240 Gy (LD_50/24 h_)	[56]
	Dehyd.	X-ray	2000 Gy (Few animals revived within 7 days)	[57]
	Dehyd.	^4^He, ^56^Fe	2000 Gy (Most animals revived within 7 days)	[57]
*Milnesium tardigradum*	Hyd.	γ-rays	5000 Gy (LD_50/48 h_)	[15]
	Dehyd.	γ-rays	4400 Gy (LD_50/48 h_)	[15]
	Hyd.	^4^He	6200 Gy (LD_50/48 h_)	[15]
	Dehyd.	^4^He	5200 Gy (LD_50/48 h_)	[15]
*Ramazzottius varieornatus*	Hyd.	^4^He	4000 Gy (Most animals survived for 48 h)	[11]
	Dehyd.	^4^He	4000 Gy (Most animals revived within 48 h)	[11]
*Hypsibius dujardini*	Hyd.	γ-rays	4180 Gy (LD_50/48 h_)	[49]
*Echiniscoides sigismundi *^2^	Hyd.	γ-rays	1270 Gy (LD_50/48 h_)	[58]
	Hyd.	γ-rays	1550 Gy (LD_50/7 days_)	[58]

^1^ Hyd.: in hydrated state; Dehyd.: in dehydrated state. ^2^ Marine tardigrade.

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
