# Peer review of "DNA Protection Protein, a Novel Mechanism of Radiation Tolerance: Lessons from Tardigrades"

_life, 2017, doi:10.3390/life7020026_

Round 1
Reviewer 1 Report
The authors present a review on the current knowledge of radiotolerance, focusing mainly on tardigrades and a novel tardigrade protein, "Dsup," involved in protecting tardigrade DNA from radiation induced damage.
The authors' original work on Dsup and this review are outstanding and important contributions to the field of radiotolerance, and more broadly abiotic stress tolerance.
Besides some minor English editing my only suggestion is that the authors include a broader discussion of proteostasis in radiotolerant organisms. For a long time researchers thought that radiotolerant organisms would be able to better protect their DNA from radiation induced damage, however for most radiotolerant organisms this has been shown not to be the case. Instead, most radiotolerant organisms have very robust mechanisms for preventing radiation induced damage to their proteomes, protecting DNA repair enzymes, etc. This allows the radiotolerant organisms to repair their DNA even after substantial damage. The authors' findings that Dsup appears to present DNA damage in the first place is very novel and important. I feel they could highlight this novelty and its importance more by expanding their discussion of the differences between tardigrade and other organisms' radiotolerant mechanisms.
Author Response
Dear Reviewer,
Thank you very much for your warm and very constructive suggestions. We completely agree with your comments.
According to your suggestions, we explain more precisely about the currently accepted mechanisms, i.e., the protection of protein through antioxidant defenses and the successive DNA repair by protected enzymes, in the abstract and the 2nd section.
And also in the last section, we summarize the review with highlighting our finding about Dsup as proposing a novel mechanism (DNA protection) for enhanced radiotolerance.
Modified text are highlighted with yellow.
Sincerely,
Takekazu
Reviewer 2 Report
Review notes
DNA protection protein, a novel mechanism of radiation tolerance: Lessons from tardigrades.
The detection of a tardigrade protein (Dsup) protecting human cells from radiation is a very important discovery that sheds light on the mechanisms of tolerance against damage caused by physical agents like radiation. This review clearly summarizes the results obtained in a previous paper (Hashimoto et al., 2016) and explains/suggests a possible mechanism pertaining to the DNA radioprotection, therefore it is considered original and interesting.
Understanding R.varieornatus Dsup protein characteristics, as well as the author’s suggestions of a possible mechanism behind its function, could benefit the direction for future studies of the protein, hopefully lead to a more clear explanation of radiation tolerance in tardigrades, and possibly have biotechnological and medical uses. The authors have clearly summarized some of their principal findings about radiation tolerance experiments of tardigrades in the lab, as well as in their previous paper. They provided the schematic structure of the Dsup protein identifying some essential parts like the nuclear localization signal (NLS), C terminal, and N Terminal regions and some of their characteristics. They also suggested a hypothetical protein from Hypsibius dujardini as a Dsup ortholog, with low identity and similarity that has two interesting conserved motifs in the NLS region. Finally they suggested a model for DNA Dsup protection by physical shielding and DNA repair, which makes sense based on the results of their previous paper, the known characteristics of this tardigrade’s unique protein and the tolerance studies to extreme conditions in tardigrades.
The authors should consider some suggestions, in particular the ones regarding D.radiodurans. Also, other minor corrections will be mentioned.
Broad comments
Pros. The authors’ review is well organized. It compiles information collected from different sources such as the general characteristics of tardigrades, and a comparison with radiotolerant organisms including prokaryotes and eukaryotes.
Line 82. Cons. The authors indicated: “Mutation of DNA repair pathways drastically compromises the radiotolerance of D. radiodurans, suggesting that their high radioresistant ability depends on powerful DNA repair system [34]”.
This statement is not entirely correct. D.radiodurans has three main factors contributing to its radiation tolerance: Cellular cleansing, antioxidant defenses, and DNA repair (Slade and Radman 2011). It has a high degree of redundancy in DNA repair and antioxidant enzymes, the bacteria is also polyploid (more copies of their genomic DNA), and it is energetic efficient when it is under stress conditions (Slade and Radman 2011).
The DNA repair system in D.radiodurans has some differences with respect to radiation-sensitive bacteria and is efficient and faithful repairing high quantities of DNA damage (Slade and Radman 2011), but my suggestion to the authors is that they either slightly expand their statement or add the words “in part” and cite additional papers with experimental results, since their reference is a review that summarizes all the factors previously mentioned. Please consider some grammatical corrections highlighted in red.
“Mutation of DNA repair pathways drastically compromises the radiotolerance of D. radiodurans, suggesting that their high radioresistance ability depends in part on a powerful DNA repair system [34, xx]”.
Line. 223.
Pros. The authors are looking to understand why a tardigrade like H.dujardini is tolerant to gamma radiation since the Dsup gene was not previously found in this species (Yoshida, et al. 2017). They found a hypothetical protein BV898_01301 as a possible orthologue of Dsup in Hypsibius dujardini, which maintains certain characteristics similar to Dsup (Hydrophobicity, charge and parts of NLS region), which is very interesting and worth testing for radioprotective properties as indicated by the authors.
Cons. As the hypothetical protein has low similarity and identity percentages, and the only highly conserved regions are short motifs not dominions, the authors could expand the examples or explanation of gene/protein diversification, duplications, loss, etc, in tardigrades species in order to provide a stronger statement for Dsup origin or evolution. They mentioned sequence diversification in intrinsically disordered proteins and an example of the LEA protein diversification, but there are not specific references that clearly show this diversification. These references should be included. I wonder if the authors searched for these probable Dsup conserved regions in other tardigrade species.
Specific comments
Line 28. “Extremophiles” are organisms that have adapted or at least are tolerant to extremely harsh environments. Most of extremophiles are single cellular organisms with simple structure, such as archaea and bacteria. However, some animals also exhibit extraordinary tolerance against extreme environments. Please add a reference
Line 31. Tardigrades are aquatic invertebrates 31 discovered by German zoologist Goeze in 1773. Their body is typically 0.1-1.2 mm length (Figure 1) and is 32 composed of five segments; one head segment and four segments having a pair of legs. Please add a reference.
Line 114. “Furthermore, tardigrades irradiated with 2,000 Gy in hydrated state can produce eggs and the eggs were able to hatch after 500 Gy of irradiation [45]”
I think this statement should be rephrased because it can be confusing to the reader. Only the eggs were irradiated in the referenced study, but the wording makes a person think the adult was irradiated, laid eggs, and then their eggs were also irradiated. Maybe there should be two references here?
Line 44. High pressure. Add the word hydrostatic in between. It will specify the kind of pressure.
In Table 1. The reference for Hypsibius dujardini DL50 is incorrect. It says 41 and the correct reference is 43
43 Beltrán-Pardo, E.; Jönsson, K.I., Harms-Ringdahl, M.; Haghdoost, S.; Wojcik, A. Tolerance to Gamma Radiation in the Tardigrade Hypsibius dujardini from Embryo to Adult Correlate Inversely with Cellular 370 Proliferation. PLoS One 2015, 10, e0133658.
Is recommended to the authors to check all the references.
Line 279. Vicente, F.; Bertolani, R. (2013) Considerations on the taxonomy of the Phylum Tardigrada. Zootaxa 2013, 279 3626, 245–248.
Author Response
Dear Reviewer,
We thank the reviewer for providing many helpful comments on our manuscript. By responding to the comments, we believe our manuscript is significantly improved, hopefully enough for publication.
We attached our precise one-to-one responses as a pdf file.
Sincerely,
Takekazu

Reviewer 3 Report
it generally good; this is certainly a
valuable review on an emerging subject, so I have no reason not to
recommend its publication.
Only a minor "typo" remark: ref 39 has a redundant Lavelle & al quotation at the end
(which is ref 40 indeed).
Author Response
Dear Reviewer,
We are glad to hear our manuscript to be worth for publication.
Thank you for pointing out our typo. We removed the redundancy as suggested.
Sincerely,
Takekazu